# Ferric Carboxymatose in Non-Hemodialysis CKD Patients: A Longitudinal Cohort Study

**DOI:** 10.3390/jcm10061322

**Published:** 2021-03-23

**Authors:** Roberto Minutolo, Patrizia Berto, Maria Elena Liberti, Nicola Peruzzu, Silvio Borrelli, Antonella Netti, Carlo Garofalo, Giuseppe Conte, Luca De Nicola, Lucia Del Vecchio, Francesco Locatelli

**Affiliations:** 1Division of Nephrology, University of Campania, Luigi Vanvitelli, 80138 Naples, Italy; m.elenaliberti@libero.it (M.E.L.); nicola.peruzzu@studenti.unicampania.it (N.P.); dott.silvioborrelli@gmail.com (S.B.); antonella.netti@studenti.unicampania.it (A.N.); carlo.garofalo@unicampania.it (C.G.); giuseppe.conte@unicampania.it (G.C.); luca.denicola@unicampania.it (L.D.N.); 2Certara Italy, 20124 Milan, Italy; patrizia.berto@certara.com; 3Department of Nephrology and Dialysis, Sant’Anna Hospital, ASST Lariana, 22042 Como, Italy; luciadelvecchio@yahoo.com; 4Department of Nephrology, Alessandro Manzoni Hospital, 23900 Lecco, Italy; Francesco.locatelli2210@outlook.it

**Keywords:** iron deficiency, anemia, ferric carboxymaltose, chronic kidney disease

## Abstract

No information is available on the efficacy of ferric carboxymaltose (FCM) in real-world CKD patients outside the hemodialysis setting. We prospectively followed 59 non-hemodialysis CKD patients with iron deficient anemia (IDA: hemoglobin <12.0/<13.5 g/dL in women/men and TSAT < 20% and/or ferritin < 100 ng/mL) who were intolerant or non-responders to oral iron. Patients received ferric carboxymaltose (FCM) (single dose of 500 mg) followed by additional doses if iron deficiency persisted. We evaluated efficacy of FCM in terms of increase of hemoglobin, ferritin, and TSAT levels. Direct and indirect costs of FCM were also analyzed in comparison with a hypothetical scenario where same amount of iron as ferric gluconate (FG) was administered intravenously. During the 24 weeks of study, 847 ± 428 mg of FCM per patient were administered. IDA improved after four weeks of FCM and remained stable thereafter. At week-24, mean change (95%CI) from baseline of hemoglobin, ferritin and TSAT were +1.16 g/dL (0.55–1.77), +104 ng/mL (40–168) and +9.5% (5.8–13.2), respectively. These changes were independent from ESA use and clinical setting (non-dialysis CKD, peritoneal dialysis and kidney transplant). Among ESA-treated patients (*n* = 24), ESA doses significantly decreased by 26% with treatment and stopped either temporarily or persistently in nine patients. FCM, compared to a FG-based scenario, was associated with a cost saving of 288 euros/patient/24 weeks. Saving was the same in ESA users/non-users. Therefore, in non-hemodialysis CKD patients, FCM effectively corrects IDA and allows remarkable cost savings in terms of societal, healthcare and patient perspective.

## 1. Introduction

Iron deficiency (ID) is common in non-dialysis CKD patients with prevalence ranging from 36% to 67%, independently from anemic status [1,2,3,4,5,6,7]. More important, the rate of clinical inertia, that is, lack of iron prescription in the presence of ID, is unacceptably high (47–86%) [1,2,3,4,5], and occurs despite International Guidelines clearly recommend offering iron supplementation as first line therapy to people with CKD-related anemia irrespective of erythropoiesis stimulating agents (ESA) use [8,9,10]. Indeed, correction of ID with iron treatment in ESA naïve patients delays the need of more expensive anti-anemic therapies (ESA or blood transfusions) [11] and, in those already receiving ESA, it improves ESA responsiveness [9]. In either condition, an initial 3-month trial with oral iron is suggested; however, in the presence of side effects and/or unsuccessful correction of ID, intravenous (IV) iron supplementation is recommended [8,9,10]. 

In Italy, available IV iron products are ferric gluconate (FG), iron sucrose and ferric carboxymaltose (FCM) [12]. The structure of FCM is constituted by a complex carbohydrate shell which tightly binds the elemental iron; the high stability of the FCM complex induces a small release of labile iron into circulation with lower risk of hypersensitivity reactions and oxidative stress [13,14]. At the same time, this pharmacological characteristic allows the administration of high doses of iron (up to 1000 mg as single administration) thus reducing the number of in-hospital visits for cumulative iron administration [12,14]. Furthermore, FCM, with ferumoxytol and iron isomaltoside (not available in Italy), are indicated by NICE guidelines as the preferred IV iron formulations in order to implement the “high-dose low-frequency” strategy indicated for CKD patients not treated by hemodialysis [8]. Nevertheless, the recent European Medicine Agency (EMA) recommendations on IV iron administration limited wider use of intravenous iron to minimize the risk of hypersensitivity reactions [15]. On the other hand, the need of resuscitation-trained staff and full emergency care facilities may pose logistic difficulties in organizing IV administration, particularly for outpatient non-dialysis CKD, with the consequence of persistent ID. In addition, the higher cost of new IV product (FCM) in comparison with older formulations (like FG) may represent a further barrier to restrict FCM prescription; however, evaluations merely based on the ex-factory price of the two compounds may be misleading, not accounting for indirect costs, such as patients’ and relatives’ loss of productivity and transportation, as well as healthcare personnel’s costs. 

We therefore performed a prospective observational study in CKD patients not receiving hemodialysis with and iron deficient anemia (IDA), intolerant or non-responders to oral iron supplementation. Primary aim was to assess the efficacy of FCM on the correction of IDA. This clinical analysis was integrated with a cost analysis to attain a more complete picture on FCM supplementation. 

## 2. Methods

We prospectively evaluated patients with non-dialysis CKD, peritoneal dialysis or with kidney transplant, consecutively enrolled in the renal clinic of University of Campania during 2019. We included patients with IDA, defined as hemoglobin (Hb) < 12 g/dL in women and <13.5 g/dL in men and TSAT < 20% or ferritin < 100 ng/mL [10], not responsive or intolerant to oral iron therapy. Diagnosis of persistent oral iron intolerance was based on patient-reported gastrointestinal disorders unchanged after switching to a different oral compound. Lack of response to oral iron was defined as TSAT and ferritin persistently below the target (<20% and <100 ng/mL, respectively), in the presence of Hb increase < 0.5 g/dL, after eight weeks of treatment. Exclusion criteria were previous episodes of hypersensitivity reactions to IV iron, pregnancy, breastfeeding, gastrointestinal blood loss, therapy with IV iron in the previous 24 weeks and hemodialysis treatment. The latter criterion was applied because hemodialysis patients usually show refractory anemia and more pronounced inflammation and high single doses of FCM (>200 mg) cannot be administered. All patients signed the informed consent for IV iron administration and use of their clinical data. The study was approved by institutional ethics committee (University of Campania L. Vanvitelli, project code 633-18).

### 2.1. Study Procedures

Study lasted 24 weeks with visits performed every four weeks. All patients received at baseline a fixed dose of 500 mg of FCM (Ferinject^®^, Vifor International, St Gallen, Switzerland) administered in 100 mL of saline solution over 15 min. We administered 500 mg instead of the higher doses recommended in the Summary of Product Characteristics of Ferinject (20 mg/kg) to avoid potentially dangerous rapid rise of Hb, particularly for patients receiving concomitant ESA therapy. For prescription of subsequent FCM doses (500 mg each at 4-week intervals), investigators used a targeted approach aimed at achieving ferritin value ≥ 100 ng/mL and TSAT ≥ 20%, unless Hb increased >2.0 g/dL in 4 weeks (conventional cut-off threshold for defining rapid rise of Hb). In the pre-specified protocol for ESA management, According to our in-center protocol for non-dialysis CKD population, ESA dose changes were decided based on measured Hb level and its change from previous visit [16]. ESA treatment is initiated when Hb is <11 g/dL in two consecutive control visits and targeted at Hb 11.0 to 12.0 g/dL. If either Hb value < 11 g/dL with an Hb increase <1 g/dL/month or Hb value ≥ 11 g/dL with an Hb decrease >2 g/dL/month, ESA dose was increased by 25%. If Hb level increased to 12–12.9 g/dL or 13–13.5 g/dL ESA dose was reduced by 25% and 50%, respectively. ESA was withdrawn if Hb > 13.5 g/dL. In this latter case, ESA treatment was resumed when Hb declined to <11 g/dL. During the study, each patient treated with ESA received always the same molecule; for ESA naïve patients, we used long-acting agents (darbepoetin or methoxy polyethylene glycol-epoetin beta), administered at initial doses of 0.45 µg/kg/week or 1.2 µg/kg/month for darbepoetin and methoxy polyethylene glycol-epoetin beta, respectively (according with summary of product characteristics). Medical history, demographics, laboratory data and current therapy, were collected at baseline. Glomerular filtration rate (eGFR) was estimated by the CKD Epidemiology Collaboration creatinine equation (CKD-EPI). At each visit, we measured in our lab hospital level of Hb, transferrin, iron and ferritin and recorded ESA doses administered. After each FCM infusion, patients were observed for the following 30 min in our clinic for occurrence of symptoms suggestive of hypersensitivity reactions [17]. Blood pressure and heart rate were recorded before and after FCM infusion; hypotension was defined as systolic BP decline > 20 mmHg or post-infusion systolic BP < 100 mmHg.

### 2.2. Efficacy Evaluation

The primary endpoint of the study was the efficacy of FCM, defined as the achievement of target for TSAT (≥20%), ferritin (≥100 ng/mL) and maintenance of Hb ≥ 11 g/dL during the evaluation period (week 20–24). For descriptive purposes, absolute changes from baseline for TSAT, ferritin and Hb, as well as their values at each time point (every four weeks), were also reported. As secondary measures of efficacy, we measured changes of ESA doses, calculated in each patient as the difference between mean ESA dose administered during the 24 weeks of the study and the respective baseline dose. We carried out subgroup analyses to compare changes of Hb, TSAT and ferritin in patients stratified by clinical setting (kidney transplant, and peritoneal dialysis non-dialysis CKD patients) and ESA use. To this aim, we calculated in each patient the absolute difference between baseline and evaluation period.

### 2.3. Economic Evaluation

An ancillary analysis was done to assess the economic impact of using FCM. We considered direct (iron and consumables for IV infusion) and indirect costs including, productivity losses (days off work due to the disease or adverse events), personnel costs and transportation costs (Table 1). All costs related to healthcare resources (e.g., iron, consumables for IV iron infusion, ESA) were estimated according to unitary costs provided by the Hospital Pharmacy. Daily loss of productivity was estimated in each patient receiving FCM infusion and cost was calculated using the per capita income in Campania region, Italy [18]. This was not applied in patients either unemployed or retired. In this latter case, we considered loss of productivity if patient was accompanied by a family member who is an active worker. Personnel costs were estimated from data of Italian Ministry of Economy by assuming a time spent for each FCM infusion of 20 min for nurses and 10 min for physicians. Furthermore, we asked each patient whether he/she attended the hospital by private, public or no transportation. Transportation costs were calculated accordingly. We compared costs associated with FCM use in our population with a hypothetical scenario in which all patients were treated with the same amount of FG administered at maximal dose allowed for single IV administration (125 mg) [12].

We also performed a sensitivity analysis by considering separately ESA treated patients to estimate also the effect on ESA dosing and costs. Changes of ESA dose (calculates as reported above) was multiplied by the cost of drug provided by the Hospital Pharmacy (Table 1). For comparison between the two iron compounds, we assumed that FG was associated with an ESA sparing effect similar to that detected with FCM. Difference between cost per patient treated with FCM versus FG represents the potential cost saving of FCM and it is calculated separately in ESA-treated and ESA-naïve patients.

### 2.4. Statistical Analysis

Continuous variables were reported as mean ± standard deviation or mean and 95% confidence interval (95%CI), according to their distribution. Achievement of target for TSAT, ferritin and Hb (primary outcome) was evaluated by McNemar test. Comparison of continuous variables across study was done by using ANOVA for repeated measures (TSAT, Hb and ESA dose) or Friedman test (ferritin). Subgroup analysis to compare absolute difference from baseline to evaluation period of TSAT, Hb and ferritin was performed by paired Student *t*-test (whole population), one-way ANOVA (among clinical settings) or unpaired Student *t*-test (between ESA users and non-users). Categorical variables, expressed as percentage, were analyzed by Cohrane’s Q-test or chi-square test. For descriptive purposes, methoxy polyethylene glycol-epoetin beta doses (µg/month) were converted to µg/week by dividing them by 4 [19]. Data were analyzed using IBM SPSS version 26 (Armonk, NY, USA). 

## 3. Results

Out of 63 eligible patients, we excluded four subjects due to cardiovascular death after week-8 visit (*n* = 1), hemodialysis start after the first FCM dose (*n* = 1), missing visits (*n* = 1) and gastrointestinal bleeding (*n* = 1). Characteristics of the 59 enrolled patients are reported in Table 2. 

Forty subjects (68%) had non-dialysis CKD, 12 (20%) had kidney transplant (KTR) and 7 (12%) were treated by peritoneal dialysis (PD). All patients were naïve for intravenous iron. During the study, patients received on average 1.7 ± 0.9 FCM infusions, corresponding to 847 ± 428 mg of iron. In particular, 48% of patients received only one infusion, 42% two infusions and 10% three or more FCM doses. No difference in FCM dose was found according to either ESA use (*p* = 0.686) or clinical setting (*p* = 0.117). 

### 3.1. Efficacy of FCM

The results on primary efficacy endpoints are reported in Table 3. 

In the evaluation period, 26 patients (44.1%) displayed a full recovery from iron deficiency by reaching both TSAT ≥20% and ferritin ≥100 ng/mL; of these, 22 patients showed a full recovery from IDA.

TSAT and ferritin levels promptly increased within the first four weeks and remained constant thereafter (Figure 1A,B). Overall, out of 354 values of ferritin and TSAT following FCM infusions, we recorded eight values of ferritin > 500 ng/mL in four patients (maximum value 743 ng/mL) and 14 values of TSAT > 40% in 10 patients (maximum value 50%). Hb levels rapidly increased up to week 8 and plateaued thereafter (Figure 1C). In the ESA-treated patients (*n* = 24), ESA dose significantly declined in the first 12 weeks and stabilized in the second half of the study (Figure 1D). In particular, ESA dose was reduced in 13 patients (54%), remained unchanged in 10 subjects (42%) and increased in one patient (4%). ESA was stopped temporarily in five subjects and persistently (up to week-24) in four patients maintaining Hb ≥ 11 g/dL; in the former subgroup, ESA was re-stared due to Hb decline to <11 g/dL but at a lower dose (−36 ± 21% versus baseline). ESA dose during week 4–24 was on average 25.5 ± 18.8 µg/week; this value was 26% lower (95%CI 9.9–42.8) than ESA dose prescribed at baseline (35.2 ± 16.8 µg/week).

Subgroup analyses are reported in Table 4. Mean increase of anemia parameters from baseline to evaluation period were similar in KTR, PD and non-dialysis CKD patients (Table 4). Similarly, changes of Hb and iron parameters were unaffected by ESA use. The full recovery from ID (TSAT ≥20% and ferritin ≥ 100 ng/mL) did not differ between these three subgroups (41.7%, 42.5% and 57.1% in KTR, PD and non-dialysis CKD patients, respectively, *p* = 0.758) as well as the full recovery from IDA (41.7%, 32.5% and 57.1% in KTR, PD and non-dialysis CKD patients, respectively, *p* = 0.434). The same held true for ESA-treated (45.8%) and -untreated patients (42.9%) (*p* = 0.821). CRP levels were similar in the three clinical settings (0.66 mg/dL (95%CI 0.09–1.23), 0.68 mg/dL (95%CI 0.15–1.22) and 0.52 mg/dL (95%CI 0.28–0.77) in KTR, PD and non-dialysis CKD patients, respectively, *p* = 0.325).

BP did not change after iron infusion (129 ± 16/76 ± 7 mmHg before FCM and 129 ± 15/75 ± 9 mmHg after FCM). No patient had systolic BP < 100 mmHg and only one patient had a drop of systolic BP >20 mmHg (from 150 to 120 mmHg) with no symptoms. No hypersensitivity reaction was recorded during the infusions and in the subsequent observation period. No patient reported symptoms after hospital discharge. Serum phosphate slightly declined from baseline to week 4 by 0.14 mg/dL (95%CI −0.08 to 0.37) (*p* = 0.23). In the following visits, serum phosphate was not regularly checked; however, among the 255 values available (72% of 354 visits after FCM), hypophosphatemia (defined as serum phosphate <2 mg/dL) was detected in only two occasions. In particular, one ND-CKD patient showed a value of 1.5 mg/dL at week 8 (at baseline phosphate was 3.0 mg/dL) and one KTR patient value of 1.9 mg/dL at week 20 (starting value was 2.3 mg/dL). Both patients were asymptomatic and FCM was administered 8 and 12 weeks before detecting low phosphate level. In non-dialysis patients, we did not observe any change in eGFR (−0.03 mL/min/1.73m^2^, 95%CI −2.62 to 2.68, *p* = 0.98) and 24h proteinuria (−0.14 g/24h, 95%CI −0.04 to 0.31, *p* = 0.20). 

### 3.2. Cost Analysis 

Cost analysis reveals that use of FCM, compared to a hypothetical scenario based on FG, would be associated with a cost saving of €288 per patient per 24 weeks, due to cost reductions related to infusion materials, personnel’s time, patient’s transportation and loss of productivity costs (Table 5). To administer the same amount of iron as FG, each patient should undergo on average 6.8 ± 3.6 FG infusions, longer stay in the clinic and more workload for physicians/nurses. We detected a significant cost saving in the FCM scenario compared to the hypothetical FG scenario, despite the higher drug cost. 

In a sensitivity analysis, we analyzed patients separately by ESA use. The reduction of ESA doses during the study (−26%) generated a saving in ESA expenditure of €211/patient/24 weeks. Implementing the same scenario with FG in this ESA-treated subgroup, even assuming that FG produced a similar ESA-sparing effect, treatment with FCM would still be cost saving. Indeed, treatment with FCM induced a cost reduction of −€33/patient/24 weeks while treatment with FG was associated with a cost excess of €250/patient/24 weeks leading to an estimated cost reduction of −€283/patient/24 weeks, similar to that reported in the whole population (Table 5). In a simulated budget impact scenario, we can estimate that the investment required to administer FG in 100 patients would allow to treat 263 patients with FCM.

## 4. Discussion

Oral iron treatment is the first-line strategy for correcting iron deficiency in CKD [8,9,10]. However, the occurrence of gastrointestinal side effects, poor patient’s adherence or inflammation-dependent impaired intestinal absorption, de facto limit effectiveness of this therapy [1,20]. According to International guidelines, patients must be in fact switched to IV iron in case of either intolerance or therapeutic failure [21]. FCM has represented a major step forward in IV iron therapy because, as compared to older agents, it allows to administer larger amounts of iron per session with lower incidence of side effects [14,22,23]. 

We found that ferritin and TSAT promptly increased in the first 4 weeks after FCM and kept constant throughout the study in the majority of patients. Response did associate with improved ESA utilization, as testified by a progressive and significant reduction of ESA doses in more than half of the 24 treated patients. 

Results in our ND-CKD subgroup are comparable with data reported in Literature. Qunibi et al. provided similar IDA correction eight weeks after FCM (Hb +1.1 g/dL, TSAT +12% and ferritin +359 ng/mL) [24]. In a longer trial lasting one year, FCM was administered with two different schedules to reach either high- or low-ferritin levels [11]. In the latter group, receiving a mean FCM dose closer to our study (1000 mg in FIND vs. 850 in our study), the entity of increase of Hb, TSAT and ferritin (+0.9 g/dL, +8.5% and +81 ng/mL, respectively) was similar to that reported in our patients (Table 4). We cannot exclude that using a more aggressive FCM dosing, aimed at reaching higher ferritin/TSAT levels, may have produced a more complete ID correction as demonstrated in the group aimed at high-ferritin of FIND trial [11]. The quantification of ESA-sparing effect of FCM in non-dialysis CKD represents a novel finding of this study. Indeed, ESA treatment was an exclusion criterion in FIND trial [11] and, in those studies in which it was allowed at baseline, no data on ESA dose change after FCM was reported [24,25]. Only one small study has reported the effect of FCM on ESA consumption [26]; however, enrolled patients were likely to be frankly resistant because they required at baseline as much as 42,000 IU/month, a value more than double of that currently administered in European patients [27]. Reduction of ESA use, therefore, may represent a further reason for implementing the use of IV iron in renal clinics in addition to the effect of FCM in delaying ESA start, as testified in FIND trial [11]. 

Data on IV iron supplementation, including FCM, are very limited outside the hemodialysis setting, particularly for PD patients and KTR. Singh et al. reported a peak Hb increase of 1.3 ± 1.1 g/dL after 1 g of IV iron sucrose (300–400 mg in multiple doses over 29 days) in an 8-week study enrolling 99 anemic PD patients of which only 18% were iron deficient [28]. In a recent small observational study, 15 PD patients receiving 400–700 mg of IV iron sucrose in divided doses showed an increase of mean Hb level (from 10.0 to 10.9 mg/dL, *p* = 0.01), mean ferritin (from 143 to 260 ng/mL. *p* = 0.005) and TSAT (from 26.2 to 33.1%, *p* = 0.07) [29]. They also reported a significant 23% reduction of ESA doses. Finally, in a single-center randomized, controlled trial, enrolling 46 PD patients, IV iron sucrose in comparison with oral iron induced after eight weeks of treatment a significant increase in mean Hb, ferritin and TSAT [30]. In our study, PD patients displayed a correction of IDA comparable with the other few studies assessing the effect of IV iron in PD patients (Table 4). Conversely, in a retrospective Spanish study enrolling 70 PD patients, Portolés-Pérez et al. reported a better correction of TSAT and ferritin (detected in 67.2% of patients after 6 month of FCM), in comparison with 42.5% in PD subgroup in our study [31]. However, the larger sample size of patients enrolled in the Spanish study, the inclusion of iron replete patients at baseline (15.7%) and the use of higher FCM doses (on average 1200 mg vs. 847 mg in our study) make difficult a direct comparison. Paucity of data in PD population likely reflects the scarce use of IV iron in this setting. Indeed, as recently reported by the Peritoneal Dialysis Outcomes and Practice Patterns Study, among 3603 PD patients from 193 facilities, only 31% received IV iron with a large variability across countries (from 55% in US to 6–17% in Australia, Canada, Japan and United Kingdom) [32].

As for PD, very few data are available in the setting of kidney transplantation. In a retrospective study, Rosen-Zvi et al. reported that in 81 kidney transplant patients, Hb increased by 1.4 g/dL (from 9.8 ± 1.4 to 11.1 ± 1.6 g/dL) and TSAT by 4.8% (from 13 ± 8% to 18 ± 10%) after three months of treatment with 800 mg of IV iron sucrose [33]. The paucity of data in this setting is surprising when considering that post-transplantation anemia, particularly in severely anemic patients, is associated with graft loss and mortality especially during the first three years [33,34,35,36]. 

Of note, the efficacy of FCM in correcting IDA was similar across the different settings examined, that is, ND-CKD, PD and KTR. This was evident in terms of absolute increase of TSAT, ferritin and Hb (Table 4) as well as when considering the rate of ID correction (41.7%, 42.5% and 57.1% in KTR, PD and non-dialysis CKD patients, respectively). Although the small sample size of KTR and PD patients does not allow a proper comparison between these settings (which was behind the scope of the present study), this finding support a more extensive use of FCM in CKD patients not on hemodialysis. On this regard, PD and KTR population is closer to ND-CKD setting than to hemodialysis population having in common a lower burden of inflammation, less blood loss, better ESA responsiveness (lower ESA doses and higher Hb levels) and residual renal function. 

These results were obtained in the absence of safety signals. Indeed, no patient reported hypersensitivity reactions and only one patient experienced a hypotensive episode (though not symptomatic). In our study, we did not recorded acute effects of FCM on renal function as well as on proteinuria levels. As to laboratory findings, ferritin and TSAT exceeded the recommended upper limit in 2.2% (8/354) and 4.0% (14/354) of recorded values and at week 24, eGFR, serum phosphate and proteinuria levels remained unchanged. Finally, we did not observed either infective episodes or need of blood transfusions. Our results are in agreement with most recent meta-analyses reporting a greater efficacy of IV iron in correcting ID and Hb levels in comparison with oral iron/placebo together with reduction in ESA use and lower blood transfusion rate in the absence of alarming signal on mortality risk, cardiovascular deaths, dialysis start and infections [37,38]. 

In parallel with the clinical efficacy, our study highlights an economic advantage of using FCM for IDA correction. The present study provides first-time evidence that in non-hemodialysis CKD patients intolerant or non-responding to oral iron, the use of FCM for correcting iron deficient anemia is effective, safe and cost saving. Indeed, we provide original evidence that FCM, in comparison with a hypothetical scenario based on FG, would be associated with cost savings of 288 euros per patient over 24 weeks of treatment. The higher cost of FCM is largely counterbalanced by the savings obtained in terms of healthcare personnel costs and patient’s productivity losses. In the sensitivity analysis limited to ESA-treated patients, the ESA-sparing effect of FCM would add a cost saving of 33 €per patient/24 weeks. 

From the healthcare services’ (hospital) perspective, these results indicate that FCM may allow a better allocation of resources that can be utilized more efficiently by dedicating economic savings and retrieved personnel’s (doctors and nurses) time to the care of more complex patients. This point is critical when considering that CKD, especially in the presence of anemia, is associated with a heavy economic burden for healthcare systems worldwide [39,40]. More important, among anemic CKD patients, the lack of specific treatment induces an additional doubling of medical expenditures mainly due to more frequent hospitalizations and emergency department visits [41,42]. We estimated that resources required to administer FG in 100 iron deficient CKD subjects would allow treating more than 260 patients with FCM. When considering the estimated number of Italian patients with CKD stage 3–5 referred to nephrology clinics (*n* = 80,301) [43,44] and the proportion of this population with ID unresponsive or intolerant to oral iron (22%) [1], it is possible to estimate yearly savings of up to €10.194.764 if patients were treated with FCM instead of FG. 

Overall, our results obtained in a real-world setting may help at overcoming the reluctance of pharmacists in acquiring FCM (mainly based on its higher costs) and of nephrologists in prescribing IV iron. This strategy, in fact, in the past years has been poorly adopted in renal clinics [1,2,3], likely because of a wrong perception of immediate harmful side effects [20,45]. More recently, the use of IV iron among options for treating ID has been further weakened by the organizational difficulties raised by the EMA recommendation requiring in-center availability of resuscitation-trained staff in a setting with full emergency care facilities [15]. This recommendation, while essential for minimizing the risk of hypersensitivity reactions, has reduced the prescription of IV iron in several settings, such as in non-dialysis patients or in renal out-patient ambulatory care settings [46]. 

The main study limitation is related to the limited number of enrolled patients; however, a very careful and strict monitoring of patients has been performed for a sufficiently long period (six months). In addition, our economic evaluation is limited to Italy that is characterized by universal coverage of healthcare system. Differences may occur with other Countries with different healthcare organization. 

In conclusion, we found that in CKD patients not on hemodialysis, FCM effectively corrects iron deficient anemia and may allow remarkable cost savings for the society, the healthcare system and the patients as well. Future studies with head-to-head comparison between FCM and FG are required to confirm superiority of FCM for the treatment of CKD patients with IDA and may provide useful data for cost-effectiveness analysis also considering patients’ quality of life.

## Figures and Tables

**Figure 1 jcm-10-01322-f001:**
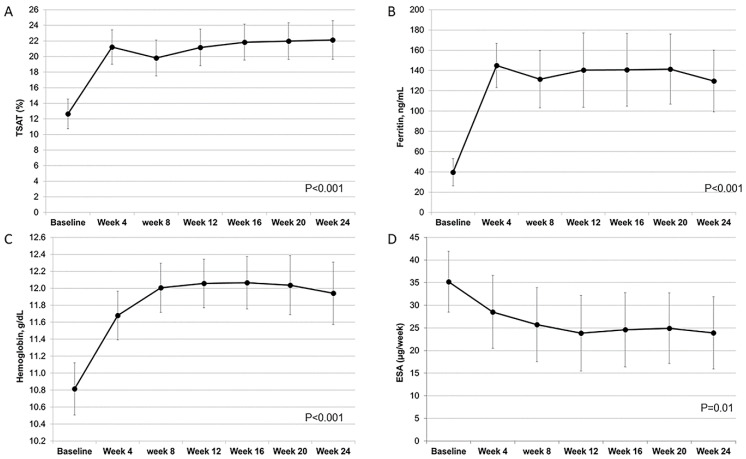
Changes of transferrin saturation (TSAT, Panel (**A**)), ferritin (Panel (**B**)), hemoglobin (Panel (**C**)) and ESA dose (Panel (**D**)) during the study.

**Table 1 jcm-10-01322-t001:** Healthcare resources and costs used for the economic analysis.

Resources	Unit Cost	Reference
Ferric gluconate 62.5 mg	€0.47	Hospital Pharmacy
Ferric carboxymaltose 500 mg	€42.64	Hospital Pharmacy
Consumables for IV iron infusion	€0.65	Hospital Pharmacy
Darbepoetin alpha (1 µg)	€1.40	Hospital Pharmacy
CERA (1 µg)	€1.49	Hospital Pharmacy
Mean annual income (Campania)	€24,732	http://dati.istat.it/Index.aspx?QueryId=22919 (accessed on 18 August 2020)
Personnel time for nurse (1 h)	€19.00	https://www.contoannuale.mef.gov.it (accessed on 18 August 2020)
Personnel time for physician (1 h)	€36.00	https://www.contoannuale.mef.gov.it (accessed on 18 August 2020)

CERA, methoxy polyethylene glycol-epoetin beta.

**Table 2 jcm-10-01322-t002:** Baseline clinical characteristics of patients.

Age (Years)	57.6 ± 17.7
Women, *n* (%)	38 (64%)
Body mass index (kg/m^2^)	26.3 ± 5.4
Diabetes Mellitus, *n* (%)	17 (29%)
History of cardiovascular disease, *n* (%)	18 (31%)
eGFR, (mL/min per 1.73 m^2^)	44.4 (35.4–53.5)
Proteinuria, (g/day)	0.75 (0.28–1.21)
Phosphorus (mg/dL)	3.8 ± 1.1
C-Reactive Protein (mg/dL)	0.57 (0.37–0.77)
Hemoglobin (g/dL)	10.8 ± 1.2
Hb < 11.0 g/dL	32 (54%)
TSAT (%)	12.7 (10.7–14.6)
TSAT < 20% *n* (%)	50 (85%)
Ferritin (ng/mL)	40 (26–53)
Ferritin < 100 ng/mL *n* (%)	56 (95%)
Darbepoetin alpha (%)	15 (25%)
Dose (µg/week)	42 ± 17
C.E.R.A. (%)	9 (15%)
Dose (µg/month)	93 ± 25

Data are mean ± SD, counts and percent or mean (95%CI).

**Table 3 jcm-10-01322-t003:** Achievement of target for transferrin saturation (TSAT), ferritin and hemoglobin at baseline and in the evaluation period (week 20–24).

	Baseline	Evaluation Period	*p*
TSAT ≥20% (%)	15.3	62.7	<0.001
Ferritin ≥100 ng/mL (%)	6.8	59.3	<0.001
Hemoglobin ≥11 g/dL (%)	45.7	83.1	<0.001

**Table 4 jcm-10-01322-t004:** Increase of TSAT, ferritin and hemoglobin from baseline to evaluation period (week 20–24) in the whole cohort and after stratification by either clinical setting or ESA use.

	TSAT (%)	Ferritin (ng/mL)	Hemoglobin (g/dL)
Overall (*n* = 59)	9.5 (5.8–13.2)	104 (40–168)	1.16 (0.55–1.77)
Clinical setting			
KTR (*n* = 12)	8.0 (3.2–12.7)	64 (4–125)	1.47 (0.71–2.23)
ND-CKD (*n* = 40)	10.3 (7.6–12.9)	114 (84–144)	1.16 (0.74–1.57)
PD (*n* = 7)	7.2 (−2.4–16.8)	55 (8–103)	1.03 (−0.37–2.43)
*p* value	0.835	0.179	0.787
ESA use			
No (*n* = 35)	9.5 (6.8–12.2)	92 (65–120)	1.45 (1.08–1.83)
Yes (*n* = 24)	9.4 (5.5–13.3)	103 (57–149)	0.84 (0.21–1.47)
*p* value	0.749	0.703	0.246

Values are mean (95%CI). KTR, kidney transplant; ND-CKD, non-dialysis CKD; PD, peritoneal dialysis.

**Table 5 jcm-10-01322-t005:** Description of costs per patient over 24 weeks associated with use of ferric carboxymaltose and costs estimated in a ferric gluconate scenario in the whole population (*n* = 59).

	Ferric Carboxymaltose	Ferric Gluconate	Difference
Drug	€72.27	€3.19	€69.08
Infusion material	€1.10	€4.41	−€3.31
Personnel costs	€20.90	€126.55	−€105.65
Transportation	€12.84	€51.34	−€38.51
Loss of productivity	€70.06	€280.22	−€210.17
TOTAL	€177.17	€465.71	−€288.54

Costs used for the economic analysis are detailed in Table 1.

## Data Availability

The data presented in this study are available on request from the corresponding author.

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
