# Peer review of "Ferric Carboxymatose in Non-Hemodialysis CKD Patients: A Longitudinal Cohort Study"

_jcm, 2021, doi:10.3390/jcm10061322_

Round 1

Reviewer 1 Report

In their longitudinal cohort study, Minutolo et. al investigate the efficacy of i.v. ferric carboxymaltose (FCM) treatment of iron deficient anemia (IDA) in a sample of 59 non- hemodialysis CKD patients from Italy, who were either not responsive or intolerant to oral iron. The efficacy was evaluated based on the increase of hemoglobin, ferritin and TSAT levels. These changes were also evaluated separately for the different clinical settings observed (PD, KTR and non-dialysis CKD) and based on ESA use (n=24). Additionally, the authors evaluated the cost of a FDM based treatment in comparison to a hypothetical ferric gluconate (FG) based treatment scenario.
To evaluate the main interest of the study (efficacy of FDM treatment), the data were analyzed using repeated measurement ANOVAs to assess the difference of Hb, Ferritin and TSAT levels at baseline and during the evaluation period (week 20-24).

The authors show significant increases of TSAT, Ferritin and Hb, while also finding a significant decrease in the need of ESA treatment during the study period (Table 3 and Fig. 1). The same comparison between baseline and evaluation period was also made depending on the different clinical settings and ESA use, where the authors did not find significant differences (Table 4). No significant changes in BP were observed after iron infusion, no hypersensitivity reaction was recorded and there were no significant changes in eGFR, serum phosphate and 24h proteinuria.
The study found a significant economic advantage of using FDM for IDA correction (in comparison to the hypothetical FG based scenario), describing an estimated cost saving of 288€ per patient over the 24 week period (Table 5).

The presentation of the data is clear. The discussion and conclusion are adequate.

I suggest the following changes:

  1. Heading:
  2. Ferric carboxymaltose is misspelled here. Please correct this.
  3. Introduction:
  4. Line 67 must be rephrased (“with and”)
  5. Methods:
  6. Please provide the name of the clinic (line 73)
  7. Did the participants receive i.v. iron prior to the start of the study period (probably, exclusion criteria states hypersensitivity)?
  8. What do the authors mean by a “higher burden of anemia” in CKD patients receiving HD, how was this defined? (prevalence, or burden as in quality of life?)
  9. Please provide the approval number/ ID of the approval from the ethics committee.
  10. ESA dose was reduced/ ESA withdrawal determined based on the Hb level of the patients. While the inclusion criteria of the study considered anemia criteria based on gender, ESA dose seems to have been administered regardless of gender. Is there a reason for this? Additionally, how many patients had to be restarted on ESA after the Hb declined <11g/dl?
  11. The authors state that ESA was administered according to the ‘in- center protocol’, what dose did the patients receive and which criteria was applied to select the n=24 patients that received ESA?
  12. Line 115: please define each time point (every four weeks?)

  13. Results:
  14. In line 168 the authors state that no difference in FCM dose could be found regarding ESA use, was there a difference depending on the clinical setting?
  15. In Table 3 and Figure 1, please state what statistical method was used to make these comparisons. ANOVA for repeated measures and t-test? Moreover, the p-values in Table 3 are based on the comparison of baseline and evaluation period, what about the p-values in the Figures?
  16. In line 173-174 the authors state the number of patients that made a full recovery from ID, what about the number of patients that made a full recovery from IDA (Hb seems to not have been observed here?). Same can be said for the group comparisons (line 191-193).
  17. In line 183, please define ‘persistently’. This refers to the Hb levels after ESA withdrawal not decreasing again <11g/dL during the 24 weeks study period?
  18. Table 4 and line 191- 195: How was this evaluated/ What statistical method was used here? ANOVA? What kind? Please specify.
  19. Cost analysis- last sentence (line 215-216): Sentence must be rephrased, because the drug cost was either way already included in the calculations. So, it does not need to “compensate” the cost. (Instead pointing out that there was a significant cost saving in the FCM scenario compared to the hypothetical FG scenario, despite the higher drug cost)
  20. Do the authors believe this hypothetical scenario is representative for the population? It seems to me like it depends a lot on the hospital location/ age of participants/ etc., as you for example included loss of productivity depending on whether they were accompanied by someone. Maybe provide more details on how this was calculated (n values)
  21. Do the authors have serum phosphorus monitored during treatment? If yes, I recommend the analysis of these data as to whether FCM treatment induces hypophosphatemia.
  22. Please add the number of injections underlying the calculation in table 5
  23. The item “loss of productivity” in the cost calculation is not uniformly true when the patient comes to the treatment without the help or need of an employed relative.
  24. Discussion:
  25. Line 242: specify, more than half of the n= 24 that received ESA, not the treated patients in general.
  26. Line 264: it seems like the ending of the sentence is missing here, please correct this.
  27. Line 273-275: the percentages given here were never mentioned in the results section of this paper. Does this refer to line 176-178? Please specify.

Reviewer 2 Report

In this paper, Minutolo and coll report the clinical efficacy of Ferric Carboxymaltose (FCM) in non-hemodialysis CKD patients, that are intolerant to oral Iron. Fifty-nine patients were treated with a single dose do FCM +/- additional doses, depending on the iron store, and were followed during a 6 month-period. The mean total dose of FCM was 847+/-428. After 6 months, 44.1% of patients displayed a full recovery from iron deficiency (Ferritin > 100 ng/ml, TSAT> 20%) which was the primary end point.

The authors performed a cost economic evaluation and showed that the FCM use was associated with a 288 euros savings as compare as a hypothetical use of Ferric gluconate (an alternative IV iron formulation) 

The subject is of interest. The study is well written but suffers majors’ flaws

1/ first it is a small study that includes different CKD profile with transplant patient, or patient with PD (N=7). I'm not sure it makes sense to include the latter as they start "dialysis" , and others reports have been published so far (eg  Portolés-Pérez PMID: 33564416)

2/ The transplant recipient group may bias the result and have major limitations. Indeed, iron homeostasis may differ from CKD patient due to difference in the local (intestinal) or general inflammation. If the iron homeostasis is restored for some transplant (low hepcidin due to low inflammation) we could hypothesize that the criteria of Iron deficiency for chronic disease (ferritin < 100 ng/ml TSAT < 20%), is not applicable, as for HD patients that become free of rHU EPO.  Data on CRP level may be useful to test his hypothesize. Finally, CKD stage may not be appropriate when assessing renal function in transplant recipient as to many factors alter Glomerular hemodynamic. 

3/ The authors claim that no information on the efficacy of FCM outside the hemodialysis setting exists... but they have mentioned the FIND CKD study that involved 616 non dialysis CKD to different regimen of FCM. It also surprising that this reference 11 is quickly mentioned in the introduction and at the end of the manuscript. Even with different regimens of FCM , the evolution of Hemoglobin, Ferritin level is quite similar , regarding the scale of the graph.

4/ The main finding that should be discussed more in details is the table 4. Basically, FCM seems to have the same effect whether it’s a CKD or transplant recipient or PD patients

5/ Data on Inflammation (CRP) Level are lacking.

6/ It seems that the authors focused too much on economic part (> 50% of the discussion section) which is not the primary nor the secondary end points. This evaluation is quite interesting but is limited as they compare the cost of FCM to the hypothetical cost of FG.

I’m surprised by the price of the darbepoetin which is twice the price we paid. Is it the same in all your country?

Minor

Line 244 “FMC” should be FMC.

Round 2

Reviewer 2 Report

No further comments. I am satisfied with the changes made by the authors

This manuscript is a resubmission of an earlier submission. The following is a list of the peer review reports and author responses from that submission.